# CARD19 Interacts with Mitochondrial Contact Site and Cristae Organizing System Constituent Proteins and Regulates Cristae Morphology

**DOI:** 10.3390/cells11071175

**Published:** 2022-03-31

**Authors:** Kariana E. Rios, Ming Zhou, Nathaniel M. Lott, Chelsi R. Beauregard, Dennis P. McDaniel, Thomas P. Conrads, Brian C. Schaefer

**Affiliations:** 1Department of Microbiology and Immunology, Uniformed Services University of the Health Sciences, Bethesda, MD 20814, USA; kariana.rios.ctr@usuhs.edu (K.E.R.); nathaniel.lott@usuhs.edu (N.M.L.); c.beauregard1@snhu.edu (C.R.B.); 2The Henry M Jackson Foundation for the Advancement of Military Medicine, Bethesda, MD 20817, USA; 3Women’s Health Integrated Research Center, Women’s Service Line, Inova Health System, Annandale, VA 22003, USA; zhoum@whirc.org (M.Z.); conrads@whirc.org (T.P.C.); 4Biomedical Instrumentation Center, Uniformed Services University of the Health Sciences, Bethesda, MD 20814, USA; dennis.mcdaniel@usuhs.edu

**Keywords:** CARD19, BinCARD, MICOS, CARD proteins, cristae, MIB

## Abstract

CARD19 is a mitochondrial protein of unknown function. While CARD19 was originally reported to regulate TCR-dependent NF-κB activation via interaction with BCL10, this function is not recapitulated ex vivo in primary murine CD8^+^ T cells. Here, we employ a combination of SIM, TEM, and confocal microscopy, along with proteinase K protection assays and proteomics approaches, to identify interacting partners of CARD19 in macrophages. Our data show that CARD19 is specifically localized to the outer mitochondrial membrane. Through deletion of functional domains, we demonstrate that both the distal C-terminus and transmembrane domain are required for mitochondrial targeting, whereas the CARD is not. Importantly, mass spectrometry analysis of 3×Myc-CARD19 immunoprecipitates reveals that CARD19 interacts with the components of the mitochondrial intermembrane bridge (MIB), consisting of mitochondrial contact site and cristae organizing system (MICOS) components MIC19, MIC25, and MIC60, and MICOS-interacting proteins SAMM50 and MTX2. These CARD19 interactions are in part dependent on a properly folded CARD. Consistent with previously reported phenotypes upon siRNA silencing of MICOS subunits, absence of CARD19 correlates with irregular cristae morphology. Based on these data, we propose that CARD19 is a previously unknown interacting partner of the MIB and the MIC19–MIC25–MIC60 MICOS subcomplex that regulates cristae morphology.

## 1. Introduction

Mitochondria are dynamic organelles that participate in a wide range of cell signaling pathways, including bioenergetics and intrinsic apoptosis. Many mitochondrial functions require tight regulation of cristae morphology, which is governed by the mitochondrial contact site and cristae organizing system (MICOS) [1,2]. Metazoan MICOS proteins assemble into two functional subcomplexes: the MIC60 subcomplex, consisting of MIC60–MIC19–MIC25 and the MIC10 subcomplex, consisting of MIC10–MIC26–MIC27, with MIC13 serving to bridge these two subcomplexes. The MIC60–MIC19–MIC25 subcomplex has, in human cell lines, been shown to associate with OMM proteins SAMM50 and Metaxins 1/2 (MTX1/2) via MIC19, thus forming the mitochondrial intermembrane bridge (MIB), which spans the intermembrane space (IMS) [2,3,4]. Interactions between the outer and inner mitochondrial membrane proteins across the MIB regulate cristae morphology via control of the formation and size of cristae junctions (CJs) [5]. Diverse human diseases have been linked to both defects in mitochondrial function [6,7,8,9,10,11,12,13,14,15,16,17,18,19,20] and in MICOS assembly and activity [21,22,23,24,25,26,27].

Caspase activation and recruitment domains (CARDs) are binding motifs commonly found in proteins that regulate immune signaling cascades and programmed cell death [28]. Although CARDs participate in signaling pathways that often require mitochondrial participation, to date, only two CARD proteins have been identified as integral mitochondrial membrane proteins: mitochondrial antiviral signaling protein (MAVS) and CARD19. Although CARD19 was initially reported to be a cytosolic and nuclear protein that regulates NF-κB activity downstream of the T cell receptor (TCR), we and others demonstrated that CARD19 is a transmembrane protein that localizes to mitochondria [4,29,30] and does not play a substantial role in TCR-directed NF-κB activation [4]. The function of CARD19 remains mostly unknown, although one study has reported a contribution of CARD19 to IFNβ and IL-6 transcription in response to antiviral signaling in certain human tumor cell lines [30].

It is well established that regulation of mitochondrial function plays a key role in macrophage differentiation, activation and survival [31]. To better understand the biological role of CARD19, particularly in macrophages, we employed an unbiased proteomics approach to identify interacting partners of CARD19 to further characterize the function of this understudied CARD protein. Here, we demonstrate that endogenous CARD19 is an outer mitochondrial membrane (OMM) protein that directly or indirectly interacts with the MICOS and MIB proteins MIC19, MIC60, MIC25, SAMM50 and MTX2. These interactions are partly regulated by the native conformation of the CARD19 CARD. *Card19*^−/−^ BMDMs and *Card19*^−/−^ fibroblasts exhibit aberrant cristae, a phenotype commonly observed upon transient knock-down of MICOS components. Taken together, these data provide strong evidence that CARD19 is a previously unknown interacting partner of specific MICOS and MIB proteins that participates in regulation of cristae morphology. 

## 2. Materials and Methods

### 2.1. Antibodies

Antibodies employed for immunoblotting and microscopy are as follows: rabbit anti-Myc (Bethyl), rabbit anti-TOMM20 (FL-145), mouse anti-β-actin (C-4), mouse anti-MAVS (E3), mouse anti-Mfn2 (XX-1), mouse anti-calnexin (E-10), mouse anti-Tim23 (H-8) (all from Santa Cruz, CA, USA), mouse anti-FLAG (M2) (Sigma-Aldrich, St. Louis, MO, USA), rabbit anti-CARD19 (HPA010990), rabbit anti-SAMM50 (HPA034537) (Atlas Antibodies, Stockholm, Sweden), mouse anti-mitofilin (2E4AD5), mouse anti-ATP synthase β (4.3E8.D10) (both from Thermo Fisher, Waltham, MA, USA), and rat anti-OLLAS ([32]; purified from hybridoma supernatant). Secondary antibodies for microscopy were goat anti-rabbit (H + L) Alexa Fluor 647, goat anti-mouse (IgG1) Alexa Fluor 488 and goat anti-mouse (IgG2a) Alexa Fluor 488 (all from Thermo Fisher, Waltham, MA, USA), used at a 1:200 dilution.

### 2.2. Mice and Cell Lines

*Card19*^−/−^ mice were generated as described previously [4]. This strain is available at The Jackson Laboratory (Stock No. 036463). To generate BMDMs, *Card19*^+/+^ and *Card19*^−/−^ mice were euthanized on day 0, followed by dissection of the tibia and pelvis. Bone marrow was flushed from the bones with 1× PBS and filtered through a 70 µm filter. Red blood cells were removed with Lymphocyte Separation Medium (Corning). An amount of 5.0 × 10^6^ cells was seeded per 100 mm Petri dishes in 30% L929 supernatant in DMEM. Half of the medium was removed and replaced with fresh 30% L929 supernatant on day 4 post-harvest. BMDMs were collected and seeded for use between days 7 and 10 post-harvest. *Card19*^+/+^ and *Card19*^−/−^ macrophages were generated from primary murine monocytes by raf-myc immortalization [33], using *Card19*^−/−^ mice [4]. Fibroblast cell lines were generated via harvest of primary fibroblasts from collagenase-digested lungs of *Card19*^+/+^ and *Card19*^−/−^ mice, followed by repeated passaging to achieve immortalization [34]. For all experiments, other than where exceptions were indicated, cells were cultured in complete DMEM containing 10% FBS, glutamine, penicillin, streptomycin, and gentamicin. Where indicated, cells were cultured in glucose-free DMEM supplemented with 10% dialyzed FBS, 4.5 g/L glutamine, 1 mM sodium pyruvate, penicillin, streptomycin, and gentamicin, and either 10 mM glucose or 10 mM galactose. Cells were passaged by trypsinization when necessary. 

Murine cDNAs encoding CARD19 [4] and MIC19 (cloned from BALB/c cDNA via PCR; identical to NCBI NM_025336) were cloned into drug-selectable retroviral vectors, with or without N-terminal epitope tags (3×Myc, 3×FLAG or 3×OLLAS). CARD19 mutants were described previously [4]. Macrophage and fibroblast cell lines were transduced with the above retroviral vectors as described [35,36], followed by selection with Zeocin or Hygromycin to generate polyclonal cell lines. 

### 2.3. Western Blotting

Cells, immunoprecipitates, or mitochondrial fractions were suspended in Laemmli buffer containing β-mercaptoethanol, followed by boiling for 5 min and shearing by sonication. Equal amounts of lysate were run on 4–20% Tris-glycine gels and transferred using a TransBlot Turbo onto nitrocellulose membranes (GE Healthcare Life Sciences, Chicago, IL, USA). Membranes were blocked with 5% milk in TBS with 0.1% Tween. Blots were incubated in block overnight at 4 °C for primary antibody or for two hours at room temperature for secondary antibody.

### 2.4. Microscopy

Cells were seeded on sterile glass coverslips, fixed with 3% paraformaldehyde/3% sucrose, permeabilized with 0.2% Triton X-100, and blocked with 10% FBS. Cells were then probed overnight with the indicated antibodies. Coverslips were imaged using a Zeiss 710 Confocal Microscope or a Zeiss LSM 980 Confocal Microscope or Zeiss ELYRA PS.1 structured illumination microscope (SIM). Images were processed using Zeiss Zen Blue or Black software. Representative insets of individual cells were selected, and single representative z-planes were exported as TIFFs. For some insets, maximum-intensity projections were generated from select z-plane ranges using ImageJ.

### 2.5. Mitochondrial Isolation and Proteinase K Protection Assays

Cell fractions were prepared using the Q proteome mitochondria isolation kit (Qiagen, Germantown, MD, USA) according to the manufacturer’s instructions. Total protein was quantified via Pierce 660 nm Protein Assay Reagent; proteinase K protection assays were performed as described in [21]. Equal concentrations of mitochondria were left untreated or treated with the indicated concentrations of proteinase K in the presence or absence of swelling buffer or Triton X-100 (1%) for 20 min on ice.

### 2.6. Immunoprecipitation and Mass Spectrometry

For all immunoprecipitation experiments, cells were lysed and washed in buffer containing 50 mM Tris-HCl, pH 7.5, 150 mM NaCl, and 1% Triton-X 100 along with commercially available cOmplete^TM^ ULTRA protease inhibitor tablets (Millipore Sigma, St. Louis, MO, USA, #5892970001) and PhosSTOP^TM^ phosphatase inhibitor tablets (Millipore Sigma, St. Louis, MO, USA, #4906845001). For mass spectrometry, lysed cells were pelleted for 10 min at 10,000 × *g* and immunoprecipitated with Myc-Trap agarose beads or negative control agarose beads (ChromoTek, Planegg-Martinsried, Germany) according to the manufacturer’s instructions. For Western blots, macrophages that were untransduced or stably expressing Myc-tagged constructs were lysed and then centrifuged for 10 min at 4 °C. Supernatants were precleared with Sepharose G beads (GE Healthcare Life Sciences, Chicago, IL, USA) for 45 min at 4 °C. 2 µg of mouse anti-Myc (9E10) was added to the pre-cleared supernatant and rotated end-over-end at 4 °C for 16 to 20 h. Sepharose Protein G bead suspensions were then added to the antibody-treated supernatant, rotated end-over-end at 4 °C for 45 min, and washed three times. Immunoprecipitates were then resuspended in 50 µL lysis/wash buffer and 50 µL 2× Laemmli buffer containing 5% *v*/*v* β-mercaptanol (Western blots) or 100 µL Laemmli buffer with 0.1 mM DTT (proteomics) and boiled for 5 min at 100 °C. The samples were then loaded and run on 4–12% NuPage Bis Tris gel (Thermo Fisher, Waltham, MA, USA). Gel bands were excised continuously throughout each entire lane. Mass spectrometry was performed on gel slices as described [37] with some differences. The Easy-Spray ion source capillary voltage and temperature were set at 2.0 kV and 275 °C, respectively. Data were searched against the Myc-CARD19 protein sequence combined with a Swiss-Prot mouse protein database (http://www.uniprot.org/uniprot/ accessed on 12 February 2020) using Proteome Discoverer (v.2.2.0.388, Thermo Fisher Scientific, Waltham, MA, USA) with the automatic decoy search option set followed by false-discovery rate processing by Percolater. Data were searched with a precursor mass tolerance of 10 ppm and a fragment ion tolerance of 0.05 daltons (Da), a maximum of two tryptic miscleavages, and dynamic modifications for oxidation (15.9949 Da) in methionine residues and phosphorylation (79.9663 Da) on serine, threonine, and tyrosine residues. Mass spectrometry raw data have been deposited in the PRIDE repository, under the project name, CARD19 Interaction Partners in Murine Macrophages, with the accession number, PXD029157.

### 2.7. Transmission Electron Microscopy

BMDMs were fixed with 2% glutaraldehyde in 0.1 M cacodylate buffer supplemented with 2 mM calcium chloride and 2 mM magnesium chloride at pH 7.4 for 2 h at room temperature. The cells were then stained with 1% osmium tetroxide and 1% uranyl acetate, dehydrated, and embedded in agar resin. Fibroblasts were prepared in the same way as BMDMs but were fixed with 2% glutaraldehyde in PBS. For serial sections, thin sections of 80 to 90 nm were cut using a Leica Ultracut UC6 ultramicrotome (Leica, Wetzlar, Germany). All images were taken with the JEOL-JEM1011 transmission electron microscope. To quantify mitochondrial deformation, at least 25 mitochondria from at least two cells were counted for each condition. Only mitochondria with cristae visible in the mitochondria were counted. Results are presented as the number of mitochondria with deformed cristae as a percent of the total mitochondria counted in each condition.

## 3. Results

### 3.1. CARD19 Is an Outer Mitochondrial Membrane Protein

To assess the subcellular localization of macrophage CARD19, we employed super-resolution structured illumination microscopy (SIM) to analyze BMDM derived from *Card19*^+/+^ mice. Endogenous CARD19 staining distributed with MAVS, an outer mitochondrial membrane protein, and ATP5F1B (ATP synthase β), an inner mitochondrial membrane protein (Figure 1A,B), forming the boundary of discrete mitochondria. Close examination of individual mitochondria revealed a punctate pattern of CARD19 that did not distribute along the entire mitochondrial structure (Figure 1A,B, insets), whereas both MAVS and ATP5F1B distributed more evenly along the entire mitochondria. Notably, CARD19 overlapped at discrete points with MAVS and generally appeared to wrap around ATP5F1B, suggesting that CARD19 may be an OMM protein. Furthermore, we employed SIM to compare CARD19 localization with calnexin (CANX), an ER marker (Appendix A). We found that while CARD19 was clearly limited to mitochondria, it sometimes appeared to make contact with calnexin-marked regions of the ER.

To better define the mitochondrial compartment to which CARD19 is localized, we performed proteinase K protection assays in isotonic buffer, hypotonic (swelling) buffer, or isotonic buffer with 1% Triton X-100. Using mitochondria isolated from primary *Card19*^−/−^ lung fibroblasts ectopically expressing 3×FLAG-CARD19, we analyzed the degradation of the samples via Western blot. Mitochondria remain intact in isotonic buffer, rendering only the OMM proteins sensitive to proteinase K degradation. However, in the presence of swelling buffer, the OMM is solubilized, generating mitoplasts with OMM, IMS, and IMM proteins susceptible to proteinase K degradation. In isotonic buffer with Triton-X 100, the entire mitochondrion is solubilized, and all mitochondrial proteins are sensitive to degradation. The degradation of the 3×FLAG-CARD19 (N-terminal FLAG tag) was compared to degradation of OMM protein Mfn2 and the IMM protein Tim23 (Figure 1C). Although a higher concentration of proteinase K was required to fully degrade 3×FLAG-CARD19, this protein was equivalently degraded in the presence or absence of swelling buffer, with slightly enhanced degradation in Triton X-100. This degradation pattern was similar to OMM protein Mfn2, but distinct from IMM protein Tim23, which exhibited enhanced degradation in hypotonic buffer. This assay was also performed with 3×FLAG-CARD19-RFP (N-terminal FLAG tag and C-terminal RFP tag), with degradation compared to the OMM protein Mfn2, the IMM protein MIC60 and the matrix protein HSP60 (Appendix A). Results were similar to Figure 1C. Additionally, the reduced degradation of the C-terminal RFP tag suggests that the CARD19 N-terminus faces the cytoplasm, whereas the C-terminus faces the intermembrane space. Overall, the proteinase K protection assay results are consistent with the interpretation that CARD19 is an OMM protein, with the CARD likely facing the cytoplasm (Figure 1C, Appendix A).

### 3.2. CARD19 Mitochondrial Localization Requires the Transmembrane Domain and Distal C-Terminus

To define the role of individual CARD19 domains in directing subcellular localization, we expressed 3×Myc-CARD19 or 3×FLAG-CARD19 (WT and domain mutants) in a *Card19*^−/−^ macrophage cell line. Using structured illumination microscopy (SIM), we imaged these macrophages after co-staining with anti-FLAG plus anti-TOMM20 (Figure 2B), or with anti-Myc plus anti-MAVS (Appendix A). 3×FLAG-CARD19 and 3×FLAG-CARD19-G73R (harboring a point mutation that disrupts the native CARD conformation [4]) each exhibited subcellular distributions highly similar to TOMM20 and MAVS (Figure 2B, Appendix A). Close examination of individual mitochondria revealed a discontinuous punctate pattern similar to the pattern observed with staining of endogenous CARD19 in BMDMs (Figure 2B inset compared with Figure 1A,B insets). The 3×FLAG-CARD19-ΔTM mutant, which contains a deletion of the CARD19 transmembrane domain [4,29], did not distribute with MAVS or TOMM20, but rather displayed nuclear and cytosolic distribution (Figure 2B, Appendix A). Likewise, 3×FLAG-CARD19-Δ130, which contains a deletion of part of the putative transmembrane domain plus the remainder of the C-terminus, was diffuse in the cytosol (Figure 2B). By contrast, the 3×Myc-CARD19-ΔCARD mutant, which contains a deletion of the entire CARD, localized to MAVS-stained mitochondria, indicating that the CARD is not required for CARD19 localization to the mitochondria (Appendix A).

By contrast to other C-terminal deletion mutants, the 3×FLAG-CARD19-Δ146 mutant, in which the C-terminus downstream of the transmembrane is deleted, exhibited a staining pattern inconsistent with cytoplasmic or mitochondrial localization (Figure 2B, Appendix A). To determine if 3×FLAG-CARD19-Δ146 localized to the endoplasmic reticulum (ER), we performed SIM on macrophages expressing 3×FLAG-CARD19, 3×FLAG-CARD19-ΔTM, and 3×FLAG-CARD19-Δ146 and co-stained with anti- calnexin, an ER marker, and anti-FLAG (Appendix A). Only 3×FLAG-CARD19-Δ146 clearly colocalized with calnexin, strongly suggesting that removal of CARD19 residues downstream of the transmembrane domain results in ER localization rather than mitochondrial localization (Appendix A). Together, these data show that both the CARD19 transmembrane domain and the distal C-terminus (aa147–183) contain required determinants of mitochondrial localization, whereas the CARD is dispensable for mitochondrial localization. 

### 3.3. CARD19 Constitutively Interacts with MICOS Complex Proteins

To identify CARD19-interacting partners in macrophages, we performed mass spectrometry (MS) analysis of immunoprecipitated 3×Myc-CARD19 expressed in *Card19*^−/−^ macrophages. We focused further analysis on candidate CARD19-interacting proteins that were significantly enriched across several experiments and that included multiple proteins previously reported to interact with each other. Interestingly, according to these criteria, the MS data showed that multiple components of MICOS were associated with 3×Myc-CARD19-WT, including MIC19, MIC25, and MIC60. Additionally, SAMM50 and MTX2, two proteins which compose part of the MIB and participate in regulation of the MIC60–MIC25–MIC19 MICOS subcomplex, were also associated with 3×MYC-CARD19-WT (Figure 3A; Appendix A). 

To confirm these results and to identify CARD19 domains required for association with MICOS components, we performed immunoprecipitation of 3×MYC-CARD19-WT and specific CARD19 mutants expressed in a *Card19*^−/−^ macrophage cell line. 3×MYC-CARD19-WT, 3×MYC-CARD19-Δ146 and 3×MYC-CARD19-ΔTM were expressed at levels similar to endogenous CARD19 in wild-type macrophages (Appendix A). However, we were unable able to confirm the relative expression of 3×MYC-CARD19-G73R and 3×MYC-CARD19-ΔCARD, likely because the CARD19 antibody specifically targets the CARD-containing N-terminus, which is mutated or deleted, respectively, in these mutants. While both MIC19 and SAMM50 efficiently coprecipitated with 3×MYC-CARD19-WT, 3×MYC-CARD19-G73R exhibited decreased association with both MICOS partners, and 3×MYC-CARD19-Δ146 and 3×MYC-CARD19-ΔTM did not exhibit detectable association with either protein (Figure 3B). Importantly, both MIC19 and SAMM50 co-immunoprecipitated with the 3×MYC-CARD19-ΔCARD mutant to a similar degree as with the full-length construct, suggesting that all necessary determinants of CARD19 association with MIC19 and SAMM50 are present in the transmembrane and C-terminal domain (Figure 3B). Additionally, none of these proteins co-immunoprecipitated with MAVS, another mitochondrial CARD protein, demonstrating the specificity of the interactions between CARD19 and MICOS components (Appendix A). Because we were unable to reliably detect MTX2 or MIC60 via IP-Western blot with existing commercial antibodies, we repeated the IP-MS employing CARD19-WT as well as the G73R, Δ146, and ΔTM mutants. Although peptide counts were low in this experiment, these data generally confirmed the trend observed in the IP-Western blot data, in which the mutants exhibited decreased association with SAMM50, MIC19, MIC25, MTX2, and MIC60 (Appendix A). Together, the above data indicate that CARD19 interacts specifically with proteins of the MIB and MIC60–MIC19-MIC25 subcomplex; these interactions do not require the CARD per se but are inhibited upon CARD misfolding.

### 3.4. Card19^−/−^ BMDMs and Lung Fibroblasts Exhibit Aberrant Cristae Morphology

Aberrant cristae morphology is a commonly observed phenotype upon siRNA silencing of MICOS proteins [3,38,39,40]. To determine if *Card19*^−/−^ mitochondria exhibit similar defects, we performed transmission electron microscopy (TEM) on *Card19*^+/+^ and *Card19*^−/−^ BMDMs and immortalized lung fibroblasts. Generally, mitochondria in BMDMs had a heterogenous morphology. However, we observed that *Card19*^−/−^ BMDMs both untreated and stimulated with LPS (a treatment known to perturb the mitochondria [41,42,43]), exhibited mitochondria with swollen cristae as well as a subset of cristae with a vesicular appearance, whereas *Card19*^+/+^ BMDM cristae were mostly lamellar (normal) (Figure 4A). Overall, 76 percent and 58.5 percent of mitochondria were normal in appearance in untreated and LPS-treated *Card19^+/+^* BMDMs, respectively (Figure 4B). In contrast, only 57.7 and 29.7 percent of mitochondria in untreated and LPS-treated *Card19*^−/−^ BMDMs, respectively, exhibited typical lamellar cristae (Figure 4B). Thus, while both *Card19^+/+^* and *Card19*^−/−^ BMDMs exhibited an increase in atypical cristae after LPS stimulation, *Card19*^−/−^ BMDMs contained proportionally more irregular cristae regardless of treatment condition. *Card19*^−/−^ fibroblasts also exhibited mitochondrial morphology characterized by swollen, vesicular cristae appearance (Appendix A). Ectopic expression of CARD19-ΔTM did not restore WT cristae morphology to *Card19*^−/−^ fibroblasts. By contrast, ectopic expression of CARD19-WT in *Card19*^−/−^ fibroblasts resulted in cristae with a more consistent lamellar appearance (Appendix A).

To further investigate the effect of CARD19 on cristae morphology, we performed TEM on *Card19^+/+^* and *Card19*^−/−^ fibroblasts. For this experiment, fibroblasts were cultured with 10 mM D-galactose for 5 days to counteract the Crabtree effect, in which cell lines rely on aerobic glycolysis and avoid oxidative phosphorylation to generate ATP [44]. Analysis of serial sections showed that *Card19^+/+^* fibroblasts had lamellar mitochondrial cristae (Figure 5), whereas *Card19*^−/−^ fibroblasts predominantly had mitochondrial cristae with an irregular and heterogenous morphology (Figure 5). Both arc-like and vesicular or stack-like cristae were apparent, along with other distorted cristae that were difficult to classify (Figure 5). Additionally, we observed that the gross morphology of the *Card19*^−/−^ mitochondria were irregular and thinner relative to their *Card19^+/+^* counterparts (Figure 5). These data demonstrate that CARD19 is an important determinant of cristae morphology in both macrophages and fibroblasts.

## 4. Discussion

CARD19 was initially reported to limit NF-κB activation downstream of TCR signaling, via a direct interaction with the CARD protein, BCL10 [45]. We have since shown that this activity and detectable interaction between CARD19 and BCL10 only occurs under conditions of substantial CARD19 and BCL10 overexpression [4]. Moreover, the initially reported CARD19 (BinCARD) ORF [45] was derived from translation of an incompletely spliced mRNA, yielding a protein with a spurious C-terminus that did not include the transmembrane domain. These early data led to further confusion regarding the subcellular localization and biological function of CARD19 [4]. Our previous data demonstrated that the single expressed isoform of CARD19 is a transmembrane mitochondrial protein in murine CD8^+^ T cells [4], a result consistent with other studies [29,30]. In this study, our analyses of murine macrophages confirm mitochondrial localization of a single isoform of CARD19. SIM imaging revealed that endogenous CARD19 and/or 3×FLAG-CARD19 exhibited a punctate staining pattern which distributed with the OMM markers MAVS and TOMM20, but appeared wrapped around the IMM marker ATP5F1B. The assignment of CARD19 to the OMM is also consistent with our biochemical data, using a proteinase K protection assay, in which CARD19 digestion was similar to the OMM control, Mfn2, but distinct from the IMM controls, Tim23 and MIC60. The reduced degradation of the C-terminal RFP tag is suggestive of an orientation in which the N-terminus faces the cytoplasm and the C-terminus faces the intermembrane space. The punctate appearance of CARD19 in our SIM studies is consistent with previous reports of super-resolution microscopy analysis of the suborganelle organization of mitochondrial proteins [46,47,48]. Moreover, stimulated emission depletion (STED) microscopy (a super-resolution technique), performed in human primary fibroblasts to visualize the endogenous MICOS components MIC60, MIC19, and SAMM50, revealed distinct protein clusters which were not evenly distributed across the mitochondria [46], similar to our SIM data regarding CARD19 distribution.

Via deletion and mutation of specific functional domains, we demonstrated that both the transmembrane domain and distal C-terminus are required for targeting of CARD19 to mitochondria. Most OMM proteins with a single membrane-spanning region have either an N-terminal (signal-anchored) or C-terminal (tail-anchored) transmembrane domain (TD). CARD19 is a member of a smaller class of OMM proteins, including TOMM22 and MIM1, which have a more centrally-located TD. In such proteins, the N-terminus faces the cytoplasm and the C-terminus faces the IMS [49]. Although the mechanisms that lead to specific targeting of such proteins to the OMM remain incompletely defined, data have shown that the cytoplasm-facing domain of such proteins is not required, and that the OMM-targeting sequences likely reside entirely within the transmembrane domain and surrounding sequences [50]. Thus, these data combined with our SIM data suggest that the CARD faces the cytoplasm. We have shown that the N-terminal (likely cytoplasmic) domain of CARD19 is dispensable for mitochondrial localization, whereas the TM and C-terminal domains are required. As our data show that deletion of the C-terminus results in ER localization of CARD19, these data suggest that a sequence required for transfer of CARD19 from the ER to the OMM is present in the C-terminal domain, similar to other proteins with a central TD [49].

Using an unbiased proteomics approach, we found that CARD19 in macrophages associates with MIB subcomplex proteins MIC60, MIC19, MIC25, SAMM50 and MTX2. Consistent with mitochondrial localization data, association with these MICOS components was highly dependent upon mitochondrial localization, with the ΔTM and Δ146 mutants showing minimal association by mass spectrometry and IP-Western blot analysis, and with the ΔCARD mutant exhibiting essentially wild-type levels of association with MIB proteins in IP-Western blot analysis. Notably, the destabilized CARD mutant G73R exhibited reduced association with SAMM50, MIC19, MIC25, and MTX2 via IP-MS analysis whereas complete deletion of the CARD did not impact association with SAMM50, MIC19 and MIC25 by IP-Western blot analysis. This seemingly contradictory observation could have several distinct explanations, including the trapping of CARD19-G73R in complexes with a spurious “partner” protein (such as a chaperone) by exposure of hydrophobic domains, due to CARD misfolding, or activation of a mitochondrial quality control mechanism that may prevent downstream association with MICOS proteins [51]. It is also possible that the CARD conformation regulates the “on/off” switch of CARD19 interactions with IMM proteins by affecting the conformation of the C terminus; when the CARD is not present, the shortened CARD19 protein is in the “interact” conformation, but the misfolding of the G73R leads to the “don’t interact” conformation of the full-length mutant CARD19. Interestingly, although the MIB has been reported to include MTX1, MTX3, and DNAJC11, we did not find evidence that CARD19 associated with these additional MIB proteins [1]. CARD19 may therefore only interact with one form of the MIB–MICOS subcomplex.

Interestingly, CARD19 has appeared in BioPlex, a mass spectrometry proteomics screen effort that employs a combination of “bait” and “prey” proteins to identify potential interacting factors of all human proteins when expressed in HEK293T and HTC116 cells [52,53,54]. In these studies, CARD19 has been reported to interact with MTX2 in both cell lines, along with MIC19 and MTX2 in HTC116 cells [52,53,54]. Interestingly, BioPlex results indicate that CARD19 also interacts with MIC27 in HEK293T cells, which forms part of the second functional MICOS subcomplex [52,53,54]. Our mass spectrometry results did not recapitulate the results of other reported interacting partners in the BioPlex studies, however; this may be due to the different cell types and species used in our experiments versus the BioPlex experiments.

CARD19 has been previously reported to regulate Type I Interferon via MAVS [30]. Specifically, this previous report indicated that HA-tagged CARD19 constitutively interacts with FLAG-tagged MAVS upon co-transfection in HEK293T cells as measured by IP-Western blot analysis in a manner independent of the CARD and dependent on the transmembrane domain of CARD19 [30]. Notably, we did not detect MAVS in association with CARD19 via either IP-MS or IP-Western blot. While it is unclear how a putative MICOS or cristae-regulating function of CARD19 reconciles with a constitutive interaction with MAVS or as a driver of IFN-β and IL-6 transcription, it is worth noting that our studies were performed largely in murine macrophages, whereas the previous study was performed in human cell lines. The discrepancy between our results and the Suzuki et al. study may be due to cell-specific functions of CARD19. Furthermore, these authors reported that siRNA knock-down of CARD19 in A549 and HEK293T cells resulted in reduced transcription of IFNβ and IL-6 upon stimulation with 3pRNA or infection with VSV [30]. We did not employ viral infection or RLR ligand stimulation as a condition in our IP-MS or IP-Western blot analyses, which may be necessary to observe MAVS-CARD19 interactions in our cell line systems [30]. Further studies will be required to reconcile our results with those reported by Suzuki et al. 

Silencing of MICOS proteins results in distorted cristae morphology; reported aberrations vary across cell types and with the specific MICOS components that have been silenced. One study reported that siRNA knock-down of MIC19 in HeLa cells led to a loss of a cristae in fifty percent of mitochondria and altered cristae in the remaining mitochondria [55]. SiRNA silencing of MIC25 in RKO and MCF-7 cells resulted in mitochondria with few if any cristae at all [56]. Furthermore, transient siRNA knock-down of MIC60 resulted in distinct cristae morphology [57]. Detailed categorization of the types of irregular cristae reported in CRISPR knock-outs of MIC19 and MIC60 in HeLa cells indicate that vesicular cristae are in fact unique to ablation of these specific MICOS constituents [40]. Other reported phenotypes of MIC19 and MIC60 knock-out in HeLa include arc-like and stacked cristae [40].

To determine if the absence of CARD19 caused changes in cristae morphology, we performed TEM analyses on WT and CARD19-deficient macrophage and fibroblast cell lines. In both types of cell lines, we found that mitochondria exhibit atypical cristae morphology compared to the wild-type morphology, with frequent vesicular forms, in the absence of CARD19 [40]. We also observed irregular cristae in *Card19*^−/−^ fibroblasts, in contrast with lamellar cristae in *Card19^+/+^* fibroblasts. Interestingly, we did not observe arc-like or stacked cristae in *Card19*^−/−^ BMDMs. This may be due to cell type and cell line specific differences in CARD19 and/or MICOS regulation. While previous work has been performed almost exclusively in human cancer cell lines (RKO, MCF-7, HeLa cells), we performed our studies exclusively in either primary murine cells or murine cell lines immortalized from primary cells. Very few MICOS studies have been performed in the context of immune cells. There is only a single report that MIC19 and MIC60 are upregulated in THP-1 macrophages upon administration of DPP8/9 inhibitor 1G244 prior to LPS stimulation; this report did not focus on MICOS complex formation and did not examine the effect of MICOS component deletion or silencing on cristae morphology [58].

To our knowledge, this is the first time that a CARD protein has been reported to interact with a MICOS subcomplex. The data we present here indicate that CARD19 interacts with MICOS–MIB complex proteins and regulates inner membrane architecture. As stated above, however, we did not examine whether CARD19 is an integral component of MICOS required for either subcomplex assembly and/or stability. Thus, we currently do not know if CARD19 is required to assemble certain MICOS subcomplexes (e.g., the MIB), or whether CARD19 is merely an interacting partner. Additionally, we did not definitively show that alteration of cristae morphology in the absence of CARD19 is the result of aberrant MICOS assembly. Future experiments should address these questions.

Indeed, cristae morphology is regulated by many distinct activities. OMA1 is reported to cleave MIC19 in response to cell stress, thus severing the MIB and disrupting cristae morphology [3]. The loss of cristae morphology occurs specifically by disrupting cristae junctions (CJ), which are reported to be reduced upon deletion or transient knock-down of various MICOS components, in particular SAMM50, MIC19, and MIC60 [3]. Furthermore, PINK1 reportedly phosphorylates MIC60 and promotes cristae junction formation in human iPSC-induced neuron lines [27]. Future experiments should address whether CARD19 plays a role in regulating cristae junction formation and size, potentially by protecting MIC19 from OMA1-mediated cleavage during stress. Beyond regulation of cristae morphology, MIC19 and MIC60 are reportedly targeted for phosphorylation by protein kinase A (PKA) [26]. Upon phosphorylation by PKA, MIC19 inhibits Parkin (PRKN) recruitment and phosphorylated MIC60 inhibits both PINK1 and PRKN recruitment upon mitochondrial membrane depolarization, thereby negatively regulating PINK1 and PRKN mediated clearance of damaged mitochondria [26]. Thus, a potential role of CARD19 in this pathway should also be investigated.

Notably, we observed only a modest loss of cristae in the absence of CARD19, and we did not measure the number or size of cristae junctions. Therefore, although our TEM data suggest that the absence of CARD19 causes aberrant cristae morphology, perhaps through effects on MICOS components, these data are inconclusive. Cristae morphology is regulated by additional proteins and processes. Prohibitins 1/2 (Phb1/2) remodel phospholipid cardiolipin to affect inner mitochondrial membrane architecture [59,60]. The transacylase Tafazzin also affects cristae morphology via cardiolipin remodeling [61]. Furthermore, ATP synthase dimerization has an impact on cristae curvature [62,63,64,65]. The GTPase OPA1 also regulates cristae morphology [66,67], and the MIC60–MIC27–MIC26–MIC10 subcomplex regulates lipid content of the IMM to regulate cristae morphology [39,68]. CARD19 may ultimately be involved in one or more of the above signaling pathways which regulate cristae architecture.

The data we present here do not address specifically whether or not CARD19 participates in signaling pathways upstream of MIB–MICOS assembly, nor what intracellular signal (if any) the N-terminal CARD may be sensing. Given that we observed via SIM that CARD19 may contact the ER, it is possible that CARD19 transmits signals from the ER to modulate cristae architecture. Interestingly, there are previous reports of preferential MICOS subcomplex formation near points of ER-mitochondria contact [69,70]. Furthermore, in a previous CARD19 study, the authors reported that the CARD of CARD19 contains three cysteines that can be fully oxidized and may therefore act as a sensor of reactive oxygen species [29]. Defects in MIC60, MIC19, MIC25, and SAMM50 have previously been linked to increased mitochondrial reactive oxygen species (mROS), alterations in mitochondrial membrane potential (MMP), and sometimes modest decreases in ATP production and oxygen consumption rate (OCR) [3,39,40,55,56,71]. It is possible that, based on the previous X-ray crystallography data, CARD19 acts as a sensor for mROS. Interestingly, databases comparing gene expression by cell type show that both human and murine *Card19* are highly transcribed in myeloid cells [72,73,74,75]. Mitochondria are key mediators of macrophage function, as macrophages undergo significant metabolic reprogramming upon activation [76,77,78,79,80]. Furthermore, mitochondrial danger signals, such as mROS and mtDNA release, reportedly directly activate pro-inflammatory machinery such as the inflammasome [41,81,82]. Given that our data show a role for CARD19 in regulation of cristae morphology, the CARD19 dependency of macrophage polarization and the functions of differentiated macrophages should be explored.

## 5. Conclusions

In this study, we report that, consistent with previous data, CARD19 is a mitochondrial membrane protein in murine macrophages and fibroblasts. Our data strongly suggest that CARD19 is an OMM protein, with the N-terminal CARD facing the cytoplasm and the C-terminus facing the intermembrane space. We demonstrated that both the transmembrane domain and the distal C-terminus are required for mitochondrial localization, whereas the N-terminal CARD is dispensable for mitochondrial targeting. Furthermore, CARD19 interacts with MICOS–MIB constituents MIC19, MIC25, MIC60, SAMM50 and MTX2. Deletion of CARD19 results in irregular cristae morphology in both fibroblasts and BMDMs, which is consistent with reported outcomes of silencing the MICOS proteins with which CARD19 interacts. Our data holistically indicate that CARD19 is a previously unidentified interacting partner of the MIB complex, which plays a role in regulation of cristae morphology. To our knowledge, this is the first report of a CARD protein interacting with MICOS and MIB proteins and regulating cristae architecture.

## Figures and Tables

**Figure 1 cells-11-01175-f001:**
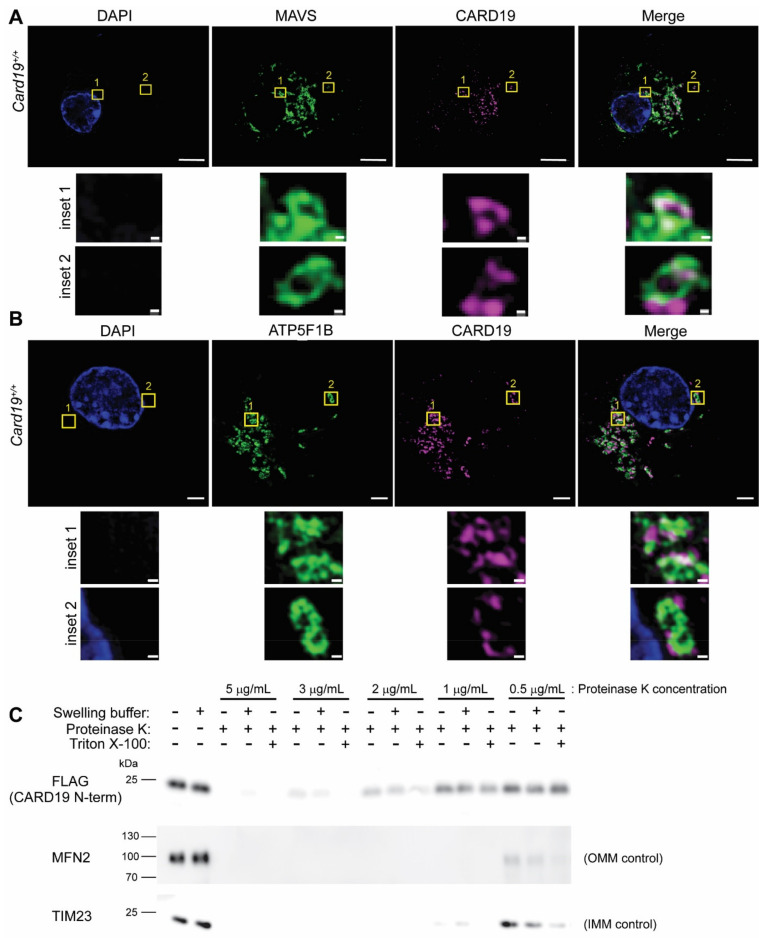
CARD19 localizes to the outer mitochondrial membrane. (**A**) BMDMs isolated from *Card19*^+/+^ mice were labeled with anti-MAVS and anti-CARD19. DAPI was included to stain nuclear DNA. SIM images were collected with an ELYRA PS.1 and processed in Zeiss Zen Black. A representative image and insets are shown. The scale bar in the image corresponds to 5 μm, and the scale bars in the inset correspond to 0.2 μm. (**B**) The same as A, but cells were stained with anti-ATP5F1B, anti-CARD19 and DAPI. (**C**) Mitochondria were isolated from a *Card19*^−/−^ fibroblast cell line stably expressing 3×FLAG-CARD19. A proteinase K protection assay was performed as a titration, using various concentrations of proteinase K, swelling buffer or Triton X-100 in the indicated lanes. Samples were separated by SDS-PAGE followed by immunoblotting with anti-FLAG to label FLAG-CARD19, the OMM marker Mfn2, and the IMM marker Tim23.

**Figure 2 cells-11-01175-f002:**
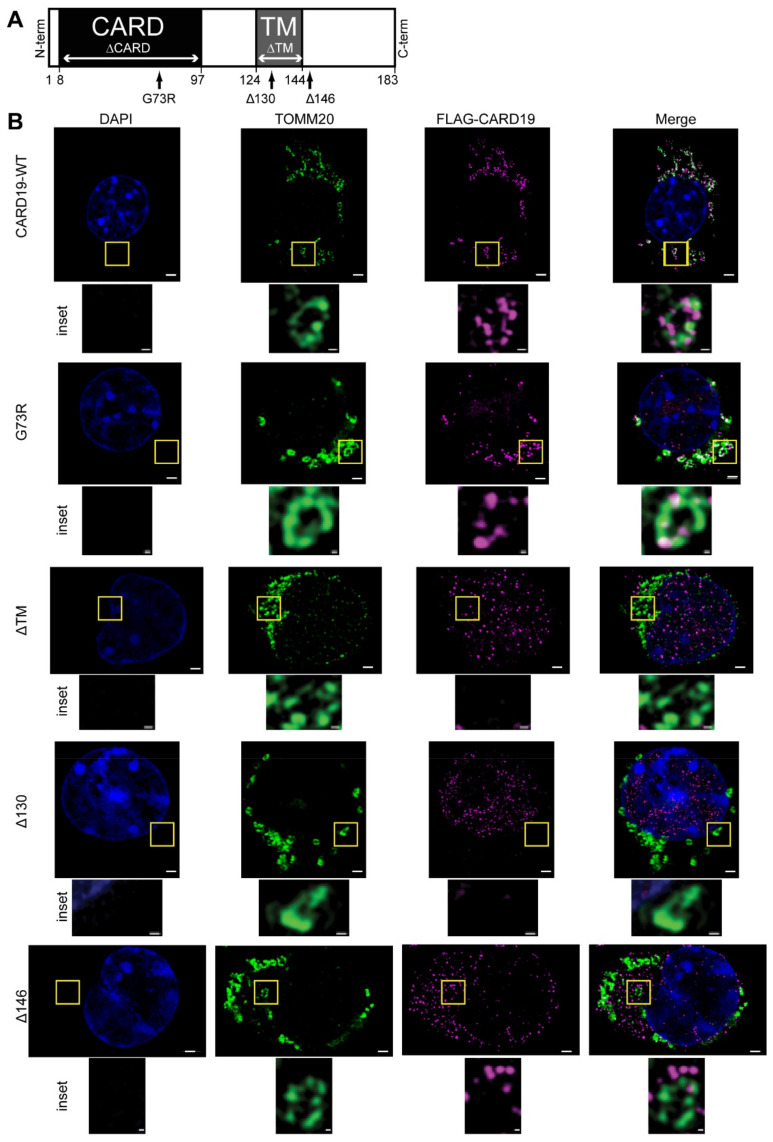
The CARD19 transmembrane domain and distal C-terminus are required for mitochondrial localization. (**A**) Diagram of CARD19 protein, with locations of domains, domain boundaries and mutations. The CARD and transmembrane domain (TM) are shaded. Point mutations and deletions are indicated by arrows. (**B**) *Card19*^−/−^ immortalized macrophages ectopically expressing FLAG-tagged WT and mutant forms of CARD19 were labeled with anti-FLAG and anti-TOMM20 and imaged via SIM. Insets contain individual mitochondria. Scale bars correspond to 5 µm in full images and 0.2 µm in insets.

**Figure 3 cells-11-01175-f003:**
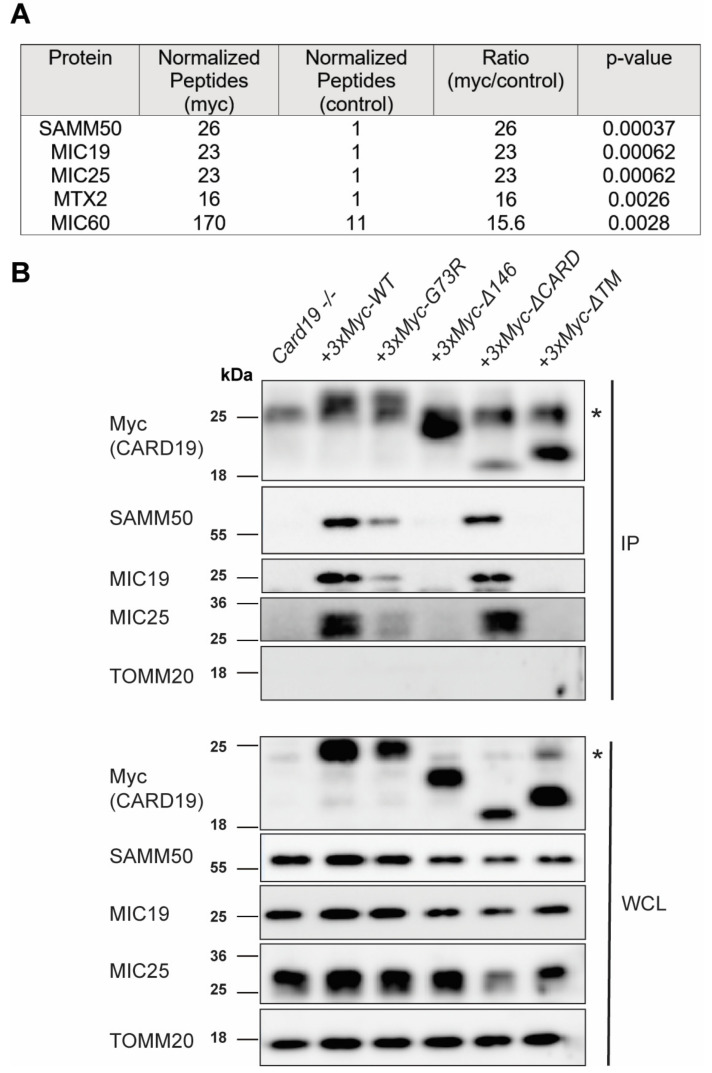
3×Myc-CARD19 interacts with MICOS and MIB proteins in macrophages. (**A**) Selected mass spectrometry analysis of immunoprecipitated proteins recovered from Myc affinity resin or control beads using lysates prepared from *Card19*^−/−^ immortalized macrophages ectopically expressing 3×Myc-CARD19-WT. See Appendix A for the complete list of significant interacting partners. (**B**) Immunoblotting analysis of anti-Myc immunoprecipitates (IP) or whole-cell lysates (WCL) prepared from *Card19*^−/−^ immortalized macrophages expressing 3×Myc-CARD19-WT. Blots were probed with antibodies specific for the indicated proteins. * Indicates the immunoprecipitating antibody light chain in the IP lanes and a non-specific 24 kDa band detected by anti-Myc in the WCL blot.

**Figure 4 cells-11-01175-f004:**
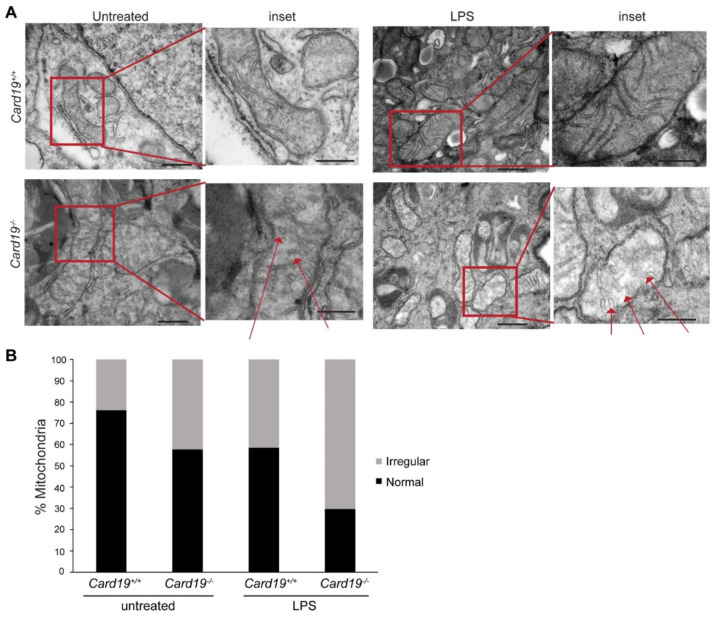
Mitochondria in *Card19*^−/−^ BMDMs have an increased proportion of irregular cristae in comparison to WT BMDMs. (**A**) BMDMs from *Card19*^+/+^ or *Card19*^−/−^ mice were either untreated or treated with LPS (100 ng/mL, 3 h), then analyzed via TEM. Insets are magnified views of mitochondria indicated by red boxes; red arrows indicate observed vesicular cristae (**B**) Mitochondria with visible cristae were counted and classified as either normal (lamellar) or irregular (swollen, vesicular). Results are presented as the percentage of either normal or irregular cristae of the total mitochondria counted for each condition.

**Figure 5 cells-11-01175-f005:**
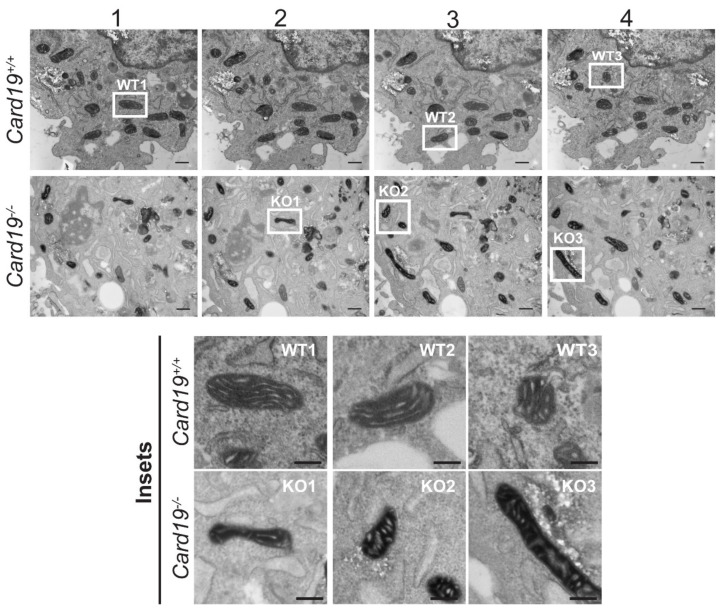
*Card19*^−/−^ fibroblasts cultured in glucose-free, galactose-supplemented medium exhibit altered cristae morphology. Immortalized lung fibroblasts from *Card19*+/+ and *Card19*^−/−^ mice were cultured for 5 days in glucose-free, galactose-supplemented DMEM, followed by serial sectioning and TEM analysis. The images shown are successive sections approximately 80–90 nm apart. Insets correspond to the respective, labeled white boxes in the full-sized images. Bar, 500 nm for full-sized images and 250 nm for insets.

## Data Availability

The mass spectrometry data presented in this study are openly available in the PRIDE repository, under the project name, CARD19 Interaction Partners in Murine Macrophages, with the accession number, PXD029157. All other data are contained with the article and Appendix A.

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
