# Peer review of "CARD19 Interacts with Mitochondrial Contact Site and Cristae Organizing System Constituent Proteins and Regulates Cristae Morphology"

_cells, 2022, doi:10.3390/cells11071175_

Round 1
Reviewer 1 Report
In this work authors intend to show a potential role for the protein CARD19. They first show results to confirm the mitochondrial localisation of the protein. Then, they identify the domain that targets it to mitochondria. The authors then show that members of both the MICOS and MIB complexes interacts with CARD19 and that the absence of the protein has implication in cristae morphology.
While the results shown in the present work helps confirming the mitochondrial localisation of CARD19 protein and gives new information about the interaction partners of the protein, no real proof that the protein is part of either of these complexes or of any functional interaction is given. While the data is relevant, some of the conclusions and several statements about the participation of CARD19 in any of this complexes or their function are not supported by the results shown.
Major points:
-Cristae morphology is not only regulated by MICOS, mentioning the other actors that are known to contribute will help to a better understanding for the reader.
-Other names for the MTXs and the MICOS CHCHDs proteins (As suggested in this 2014 Pfanner’s paper: Uniform nomenclature for the mitochondrial contact site and cristae organizing system).
-The authors mention that Card19 localises to the OMM by showing overlapping of CARD19 with the outer membrane protein MAVS and the inner membrane ATPF5B, but the images do not show a significant overlap between CARD19 and the mitochondrial proteins, and CARD19 appears to be in proximity to mitochondria. The authors then support their claim that CARD19 localises to the OMM using a proteinase K digestion assay. However, they mention that the RFP tag is not fully digested by proteinase K, as opposite to the FLAG tag, because the C-terminus faces the IMS, but no protein from the IMS is used as a control.
In the same figure, Triton X-100 is a detergent that exposes all the protein content to the proteinase K, but the digestion of the RFP tag Card19 is the same with or without Triton X-100, even in the presence of swelling buffer. Can the authors explain why is that happening? Furthermore, can the authors please clarify if by swelling buffer, they refer to swelling mitochondria to disrupt the outer membrane and obtain two fractions; the IMS exposed and mitoplasts. If this is the case, what is the fraction that they are showing?
-In Figure 3, the authors mention that MIC19 and SAMM50 co-immunoprecipitated with 3xMYC-CARD19-ΔCARD19 construct to a similar degree as with the full construct. However, immediately after they write: “Because we were unable to reliably detect MTX2 or IMMT via IP-western blotting with existing commercial antibodies, we repeated the IP MS employing WT CARD19 as well as the G73R, Δ146, and ΔTM CARD19 mutants. These data confirmed the trend observed in the western blot IP data, in which the mutants exhibited decreased association with SAMM50, CHCHD3, CHCHD6, MTX2, and IMMT” but the CARD19 G73R mutant has “a point mutation that disrupts the native CARD conformation”. In the discussion, authors mention that this can be due to trapping of this mutant in complexes. But the CARD mutant can be targeted to mitochondria and sorted into the OMM similarly to wt CARD19, suggesting that such complexes are not formed, or not in a way that affects its insertion. So, if the protein is properly inserted in the OM and the CARD domain is exposed to the cytosol, how can this affects the interaction of the IMS C-terminal domain with the proposed interacting proteins?
Can the authors explain better this point about why does the absence of the CARD19 domain have no impact the potential interaction of the protein with SAMM50 and MIC19, whereas a point mutation that disrupts the CARD domain does?
-In Figure 4, authors show representative TEM images to support that when CARD19 is not present, cristae are disrupted like when MICOS members are silenced. Can the authors provide the quantification of their set of images and mention in the text the number of cells or mitochondria that such TEM images represent? Furthermore, since the authors mentioned that the effect of the absence of CARD19 is similar to that of MICOS, and in the discussion they talked about the differences in cell lines, providing a control in where the authors silence components of the MICOS complex to compare with the effect of CARD19 will help to support their claim.
-In the same section, the authors conclude that the TEM images demonstrate that CARD19 is an important determinant of cristae morphology in macrophages and fibroblasts. Authors only show this single experiment supporting this claim which makes their conclusion overstated. Comparing the effects seen in their CARD19 -/- with the so far known regulators of cristae morphology will be useful to support such conclusion.
-On line 355 the authors stated that CARD19 is a member of the MICOS complex, but they only showed co-IP and the MS results. While such results can give a hint about the proximity and potential interaction between proteins, it does not allow to conclude that CARD19 is part of the actual MICOS complex or the MIB complex. In particular, since both SAMM50 and MTX2 are OMM proteins, so they are likely to localise in vicinity to CARD19, if it is also an OMM protein. Furthermore, no evidence that CARD19 is present in the >1000 MIB complex is present whatsoever. BNs of the protein showing that it is present in the same MIB complex as SAMM50 are needed to reach to such a conclusion.
Minor:
-Reference for: Interactions between the outer and inner mitochondrial membrane proteins across the MIB regulate cristae morphology via control of the formation and size of cristae junctions (CJs).
- Rephrase: Here, we demonstrate that endogenous CARD19 is an outer mitochondrial membrane (OMM) protein that directly or indirectly interacts with the MICOS proteins CHCHD3, IMMT, CHCHD6, SAMM50 and MTX2 in a manner partly regulated by the native conformation of the CARD19 CARD
-Line 250: also identified in the as associated with 3×MYC-CARD19-FL
-Line 430/431: “The role of CARD19 in regulating the MIB and therefore cristae morphology remains unclear.” In this sentence, the authors are assuming that CARD19 regulates MIB and cristae, but, as mentioned before this is an overstatement as not enough evidence of any regulation from CARD19 to neither the formation nor the stability of MIB or MICOS complexes is provided in this work.
Author Response
Rev 1
Comments and Suggestions for Authors
General) In this work authors intend to show a potential role for the protein CARD19. They first show results to confirm the mitochondrial localisation of the protein. Then, they identify the domain that targets it to mitochondria. The authors then show that members of both the MICOS and MIB complexes interacts with CARD19 and that the absence of the protein has implication in cristae morphology.
While the results shown in the present work helps confirming the mitochondrial localisation of CARD19 protein and gives new information about the interaction partners of the protein, no real proof that the protein is part of either of these complexes or of any functional interaction is given. While the data is relevant, some of the conclusions and several statements about the participation of CARD19 in any of this complexes or their function are not supported by the results shown.
RESPONSE): In line with the overall critique of Rev 1, we have moderated some of our conclusions relating to the interaction between CARD19 and MICOS. In particular, we no longer refer to CARD19 as a MICOS component, but rather as a MICOS-associated protein.
Major points:
-1) Cristae morphology is not only regulated by MICOS, mentioning the other actors that are known to contribute will help to a better understanding for the reader.
-Other names for the MTXs and the MICOS CHCHDs proteins (As suggested in this 2014 Pfanner’s paper: Uniform nomenclature for the mitochondrial contact site and cristae organizing system).
RESPONSE): We have added a paragraph containing information on other regulators of cristae morphology. The names for MICOS proteins have been changed, in line with Pfanner’s paper.
-2) The authors mention that Card19 localises to the OMM by showing overlapping of CARD19 with the outer membrane protein MAVS and the inner membrane ATPF5B, but the images do not show a significant overlap between CARD19 and the mitochondrial proteins, and CARD19 appears to be in proximity to mitochondria. The authors then support their claim that CARD19 localises to the OMM using a proteinase K digestion assay. However, they mention that the RFP tag is not fully digested by proteinase K, as opposite to the FLAG tag, because the C-terminus faces the IMS, but no protein from the IMS is used as a control.
RESPONSE): To clarify, we do not state that there is overlap between the distinct mitochondria proteins in the SIM data of figure 1. Rather, we state that CARD19 distributes with other OMM mitochondrial markers – i.e, it appears to be in the same region defined by other (punctate) OMM markers, but distinct from the inner membrane marker, ATPF5B. Because SIM resolution is improved over the hypothetical maximum resolution of confocal microscopes, we can, to some degree, visualize discrete localization patterns of each protein, rather than the general overlap you’d see with confocal. Other super-resolution studies of mitochondrial proteins show similar discrete clusters of OMM proteins that often do not overlap with other OMM proteins (see PMID 20205711, 21799113, 23676277, all of which are cited in our Discussion). With regard to the ProK assay, we agree with both reviewers that there were problems with assay. As detailed in points below, we have repeated this assay with a version of CARD19 tagged on the N-terminus ONLY (due to technical problems with a FLAG- and Myc-dual-tagged CARD19 construct). We believe the aggregate data strongly support our conclusion that CARD19 is most likely an OMM protein, with the CARD facing the cytosol, but we have made the conclusion regarding orientation less emphatic, since we were unable to assess digestion of the C-terminal domain.
3) In the same figure, Triton X-100 is a detergent that exposes all the protein content to the proteinase K, but the digestion of the RFP tag Card19 is the same with or without Triton X-100, even in the presence of swelling buffer. Can the authors explain why is that happening? Furthermore, can the authors please clarify if by swelling buffer, they refer to swelling mitochondria to disrupt the outer membrane and obtain two fractions; the IMS exposed and mitoplasts. If this is the case, what is the fraction that they are showing?
RESPONSE): Because both reviewers identified clear issues with the ProK assay, we have repeated the assay with several changes. Firstly, we performed the experiment as a titration, as we found that different proteins have somewhat differing sensitivities to ProK (presumably reflecting the degree of association with partner proteins, glycosylation, accessible domains for cleavage, etc); in particular, a portion of FLAG-CARD19 is highly resistant to ProK cleavage (this may also, in part, reflect the exquisite sensitivity of the M2 anti-FLAG monoclonal Ab). AS mentioned above, although we attempted to use a dual tagged CARD19 construct with N-terminal FLAG and C-terminal Myc, we found that the C-terminal Myc tag prevented mitochondrial localization. Because of this issue, we moved forward with a version of CARD19 with only an N-terminal FLAG tag (which unfortunately did not allow us to assess the degradation of the C-terminal domain). We assessed degradation of FLAG-CARD19 vs. the OMM marker Mfn-2 and the IMM marker Tim-23. FLAG-CARD19 is equivalently degraded across all three conditions (with slightly enhanced degradation in Triton), as is Mfn2, consistent with both proteins being in the OMM. By contrast, the IMM protein, Tim23, exhibits enhanced digestion in swelling buffer.
Note that swelling buffer is a hypotonic buffer, which causes the dissociation of the OMM, thus creating mitoplasts, which are comprised only of IMM and matrix. IMS and OMM proteins are in the suspension of the sample, rather than as part of the mitoplasts. Thus, IMM proteins are rendered more susceptible to ProK degradation after swelling buffer treatment. Each lane represents an independent sample (and not fractions collected).
-4) In Figure 3, the authors mention that MIC19 and SAMM50 co-immunoprecipitated with 3xMYC-CARD19-ΔCARD19 construct to a similar degree as with the full construct. However, immediately after they write: “Because we were unable to reliably detect MTX2 or IMMT via IP-western blotting with existing commercial antibodies, we repeated the IP MS employing WT CARD19 as well as the G73R, Δ146, and ΔTM CARD19 mutants. These data confirmed the trend observed in the western blot IP data, in which the mutants exhibited decreased association with SAMM50, CHCHD3, CHCHD6, MTX2, and IMMT” but the CARD19 G73R mutant has “a point mutation that disrupts the native CARD conformation”. In the discussion, authors mention that this can be due to trapping of this mutant in complexes. But the CARD mutant can be targeted to mitochondria and sorted into the OMM similarly to wt CARD19, suggesting that such complexes are not formed, or not in a way that affects its insertion. So, if the protein is properly inserted in the OM and the CARD domain is exposed to the cytosol, how can this affects the interaction of the IMS C-terminal domain with the proposed interacting proteins?
Can the authors explain better this point about why does the absence of the CARD19 domain have no impact the potential interaction of the protein with SAMM50 and MIC19, whereas a point mutation that disrupts the CARD domain does?
RESPONSE): With regard to the IP-immunoblotting data and the IP-mass spec data, the protocols are not identical. In the IP-mass spec experiments, a resin-coupled myc nanobody was used. In the IP-immunoblotting experiments, lysates were first incubated with the anti-Myc mAb 9E10, followed by immunoprecipitation with protein G Sepharose. Thus, protein complexes differing in stability might be differentially recovered in the two methods, given differences in shear force, number of washes, nanobody vs. bivalent IgG, etc. We put more faith in the IP-immunoblotting experiment, as the mass spec experiment with the Card19 mutants had low peptide counts for most of the MICOS proteins (for this reason, this mass spec experiment was placed in the supplement). We agree that the behavior of the different mutants in the IP-immunoblotting experiment (particularly dCARD vs G73R) is rather surprising. However, this result has been quite consistent across IP-western experiments, and we are thus confident in the data – we are presenting the result of this experiment and offering the simplest interpretation that we can envision. The best potential explanation that we can offer is that the misfolded CARD19-G73R has disrupted localization, due to spurious interactions between the misfolded CARD (perhaps due to exposure of hydrophobic domains) and other (non-MICOS) proteins, perhaps including mitochondrial chaperones. It is reasonable to propose that such mis-localization may not occur in the complete absence of the CARD, due to the absence of the misfolded G73R CARD that may direct such interactions. Although we can offer no proof for this model, we do believe that it is plausible. We have tried not to oversell our interpretation of these data.
-In Figure 4, authors show representative TEM images to support that when CARD19 is not present, cristae are disrupted like when MICOS members are silenced. Can the authors provide the quantification of their set of images and mention in the text the number of cells or mitochondria that such TEM images represent? Furthermore, since the authors mentioned that the effect of the absence of CARD19 is similar to that of MICOS, and in the discussion they talked about the differences in cell lines, providing a control in where the authors silence components of the MICOS complex to compare with the effect of CARD19 will help to support their claim.
RESPONSE): Quantification was added to Figure 4; see 4B. At least 25 mitochondria with visible cristae across 1-2 cell in each condition were counted. Cristae were considered normal if lamellar. Irregular cristae were swollen or vesicular non-lamellar cristae. Normal or irregular cristae are tabulated as a percentage of total mitochondria counted in each sample. In each case, there is a lower percentage of mitochondria having cristae with normal morphology. Given that these differences are not dramatic, we have described the effect of CARD19 on cristae morphology as “not definitive.”
-5) In the same section, the authors conclude that the TEM images demonstrate that CARD19 is an important determinant of cristae morphology in macrophages and fibroblasts. Authors only show this single experiment supporting this claim which makes their conclusion overstated. Comparing the effects seen in their CARD19 -/- with the so far known regulators of cristae morphology will be useful to support such conclusion.
RESPONSE): We agree that our data are not sufficient to strongly demonstrate that the alteration of cristae morphology in the absence of CARD19 is directly due to MICOS deficiency. As stated in response to the previous critique, we have changed the text to reflect this and have softened the language of our conclusions. We have also provided quantification of our observations of vesicular morphology of the BMDMs in Fig. 4A
-6) On line 355 the authors stated that CARD19 is a member of the MICOS complex, but they only showed co-IP and the MS results. While such results can give a hint about the proximity and potential interaction between proteins, it does not allow to conclude that CARD19 is part of the actual MICOS complex or the MIB complex. In particular, since both SAMM50 and MTX2 are OMM proteins, so they are likely to localise in vicinity to CARD19, if it is also an OMM protein. Furthermore, no evidence that CARD19 is present in the >1000 MIB complex is present whatsoever. BNs of the protein showing that it is present in the same MIB complex as SAMM50 are needed to reach to such a conclusion.
RESPONSE): We agree with the reviewer assessment and we have changed the language of the Discussion, Title, etc., to reflect the fact that we have shown a biochemical association between CARD19 and MICOS components (rather than CARD19 being a member of the MICOS complex). We would also emphasize out that we have included non-MICOS OMM proteins in our IP data (e.g., TOMM20 in Fig 3B, MAVS in Supp. Fig 4B), to show that the co-precipitation with MICOS components is quite selective.
Minor:
-1) Reference for: Interactions between the outer and inner mitochondrial membrane proteins across the MIB regulate cristae morphology via control of the formation and size of cristae junctions (CJs).
RESPONSE): Reference to an appropriate review has been added.
- 2) Rephrase: Here, we demonstrate that endogenous CARD19 is an outer mitochondrial membrane (OMM) protein that directly or indirectly interacts with the MICOS proteins CHCHD3, IMMT, CHCHD6, SAMM50 and MTX2 in a manner partly regulated by the native conformation of the CARD19 CARD
RESPONSE): We have rephrased this sentence as follows: “Here, we demonstrate that endogenous CARD19 is an outer mitochondrial membrane (OMM) protein that directly or indirectly interacts with the MICOS proteins MIC19, MIC60, MIC25, SAMM50 and MTX2. These interactions are partly regulated by the native conformation of the CARD19 CARD. Card19-/- BMDMs and Card19-/- fibroblasts exhibit aberrant cristae, a phenotype commonly observed upon transient knock-down of MICOS components.”
-3) Line 250: also identified in the as associated with 3×MYC-CARD19-FL
RESPONSE): Thank you for pointing out this error. We have corrected this sentence. Also, for uniformity, all instances of “FL” have now been replaced by “WT”
-4) Line 430/431: “The role of CARD19 in regulating the MIB and therefore cristae morphology remains unclear.” In this sentence, the authors are assuming that CARD19 regulates MIB and cristae, but, as mentioned before this is an overstatement as not enough evidence of any regulation from CARD19 to neither the formation nor the stability of MIB or MICOS complexes is provided in this work.
RESPONSE): We have removed this sentence and limited our conclusions to appropriately reflect that CARD19 may regulate cristae morphology independently of MICOS regulation.

Reviewer 2 Report
In their manuscript “CARD19 is a mitochondrial contact site and cristae organizing system constituent that regulates cristae morphology”, Rios et al. investigate the role of CARD19 in mitochondrial morphology and interactions with the MICOS complex. The study reports that CARD19 localizes to mitochondria and interacts with a subcomplex of MICOS, namely the IMMT-CHCHD3-CHCHD6 subcomplex, as well as with SAMM50 which form together the MIB (mitochondrial bridging complex). Furthermore their results indicate that the transmembrane domain and the C-terminus are required for mitochondrial localization of CARD19. Rios et al. claim that the C-terminus is facing the intermembrane space and is the interaction site with the MIB. Intriguingly, CARD19 has a role in mitochondrial inner membrane morphology which the authors show clearly by electron micrographs. The authors suggest this function of CARD19 may be related to its interaction with the MIB.
This study reports very interesting findings and the microscopy data regarding CARD19 localization and cristae architecture are convincing. However there are several issues that must be addressed in order to provide all relevant information and to be able to draw robust conclusions.
Major points:
- The proteinase K protection assay does not clarify the topology of CARD19 in the mitochondrial outer membrane since thet tags on both termini are degraded equally well by protease treatment without swelling etc. The authors conclude from the persistence of a small amount of the C-terminal RFP tag that the C-terminus is protected from degradation, but this small level of undegraded RFP persists also after swelling or Triton treatment with PK and therefore probably corresponds to small amounts of aggregated RFP. Thus the experiment shows surface-exposure of both termini, which clearly does not make sense for an integral membrane protein. The authors need to provide experimental proof for the proposed topology with the C-terminus inside the IMS. It is possible that the large RFP tag interferes with CARD19 membrane insertion, so the authors could perform the PK assay for example with CARD19 variants separately tagged at N- or C-terminus with a small peptide tag. Additionally the mitochondrial isolation protocol should be optimized since the controls IMMT and HSP60 reveal some issues with the integrity of the membranes (significant degradation of internal proteins with PK in absence of swelling or detergent).
- The localization of most CARD19 variants to foci in the periphery of mitochondria and the finding that the C-terminal deletion mutant localizes to the ER suggests that CARD19 may be localized to mitochondria-ER contact sites. Therefore it is crucial to include microscopy data showing CARD19 localization along with mitochondrial and ER marker proteins for the same cells, not separately. The authors should do this at least for full-length CARD19, the ∆146 mutant that localizes to the ER, and the ∆CARD mutant. If the topology proposed by the authors is correct, the CARD domain is exposed on the mitochondrial surface and could be involved in protein-protein interactions at organelle contact sites.
- The presentation of the mass spectrometry results is unacceptable. The authors need to provide an unbiased list of interacting proteins sorted by enrichment and providing the actual p-values, not stars. Please refer to published articles for reference. As the data are presented now it is impossible to assess if MICOS / MIB components are the only or the most abundant interactors or rather further down the list. It would be interesting to see whether any ER proteins are present among the interactors.
- The authors show that in the absence of CARD19, mitochondrial inner membrane architecture is altered. While the data are convincing, they only demonstrate swelling / deformation of cristae, not a loss of crista junctions and/or cristae as in MICOS deletion mutants. Therefore, the data do not support the conclusion that CARD19 is a constituent of MICOS (even more so in light of the previous points). This work does not even show that MICOS is altered in the absence of CARD19. Membrane alterations can be the result of dysregulation in any number of processes (many of which take place at organelle contact sites). Thus the authors should limit their conclusion and title to the demonstrated findings that CARD19 interacts with MICOS components (among other proteins?) and has a role in inner membrane architecture.
Minor points:
- The microscopy images in Fig. 1,2 and SFig. 1,2 are very small and often quite dark, and in many cases it isn't clear immediately to which parts of the image insets correspond. Please modify accordingly. Also for a better comparison an inset of the untreated mitochondria in Fig. 4A would be helpful.
- In Fig. 3B CARD19 is re-expressed into the CARD19-/- cells and claimed to express to a normal WT level. It would be more scientifically correct to show a figure to underline this claim either in the manuscript or in the supplementary files.
- The authors should proofread their manuscript and check that the mentioned figures in the text are linked to the corresponding figures displayed and that figure legends are correct.
Author Response
Rev 2
General) In their manuscript “CARD19 is a mitochondrial contact site and cristae organizing system constituent that regulates cristae morphology”, Rios et al. investigate the role of CARD19 in mitochondrial morphology and interactions with the MICOS complex. The study reports that CARD19 localizes to mitochondria and interacts with a subcomplex of MICOS, namely the IMMT-CHCHD3-CHCHD6 subcomplex, as well as with SAMM50 which form together the MIB (mitochondrial bridging complex). Furthermore their results indicate that the transmembrane domain and the C-terminus are required for mitochondrial localization of CARD19. Rios et al. claim that the C-terminus is facing the intermembrane space and is the interaction site with the MIB. Intriguingly, CARD19 has a role in mitochondrial inner membrane morphology which the authors show clearly by electron micrographs. The authors suggest this function of CARD19 may be related to its interaction with the MIB.
This study reports very interesting findings and the microscopy data regarding CARD19 localization and cristae architecture are convincing. However there are several issues that must be addressed in order to provide all relevant information and to be able to draw robust conclusions.
Major points:
- The proteinase K protection assay does not clarify the topology of CARD19 in the mitochondrial outer membrane since the tags on both termini are degraded equally well by protease treatment without swelling etc. The authors conclude from the persistence of a small amount of the C-terminal RFP tag that the C-terminus is protected from degradation, but this small level of undegraded RFP persists also after swelling or Triton treatment with PK and therefore probably corresponds to small amounts of aggregated RFP. Thus the experiment shows surface-exposure of both termini, which clearly does not make sense for an integral membrane protein. The authors need to provide experimental proof for the proposed topology with the C-terminus inside the IMS. It is possible that the large RFP tag interferes with CARD19 membrane insertion, so the authors could perform the PK assay for example with CARD19 variants separately tagged at N- or C-terminus with a small peptide tag. Additionally the mitochondrial isolation protocol should be optimized since the controls IMMT and HSP60 reveal some issues with the integrity of the membranes (significant degradation of internal proteins with PK in absence of swelling or detergent).
RESPONSE): Please see response to Rev 1. We have found this assay quite frustrating, as the sensitivity of distinct proteins to ProK differs markedly. FLAG-CARD19, in particular, is quite resistant to ProK digestion. We do not agree that these issues reflect integrity of the mitochondria. Also, we were unable to produce a version of CARD19 with both N- and C-terminal peptide tags that was successfully incorporated into mitochondria, and we therefore used a CARD19 construct having an N-terminal FLAG and no C-terminal tag. We have presented the data as a titration, in order to demonstrate that there is a range of ProK concentrations, specific to each mitochondrial protein, that yields an appropriate ProK sensitivity pattern. The data we present are consistent with CARD19 being an OMM protein (including the super-resolution microscopy data). The likely orientation is with the CARD facing the cytosol, given what is known about mitochondrial import of this type of transmembrane protein into mitochondria. We have been careful not to overstate the conclusions regarding orientation, however.
- The localization of most CARD19 variants to foci in the periphery of mitochondria and the finding that the C-terminal deletion mutant localizes to the ER suggests that CARD19 may be localized to mitochondria-ER contact sites. Therefore it is crucial to include microscopy data showing CARD19 localization along with mitochondrial and ER marker proteins for the same cells, not separately. The authors should do this at least for full-length CARD19, the ∆146 mutant that localizes to the ER, and the ∆CARD mutant. If the topology proposed by the authors is correct, the CARD domain is exposed on the mitochondrial surface and could be involved in protein-protein interactions at organelle contact sites.
RESPONSE): We agree that CARD19 could very well localize to ER-mitochondrial contact sites. We have included SIM microscopy of BMDMs labeled with anti-CARD19 and anti-Calnexin, which demonstrates that Card19 sometimes makes contact with the ER. See new Supplemental Figure 1.
- The presentation of the mass spectrometry results is unacceptable. The authors need to provide an unbiased list of interacting proteins sorted by enrichment and providing the actual p-values, not stars. Please refer to published articles for reference. As the data are presented now it is impossible to assess if MICOS / MIB components are the only or the most abundant interactors or rather further down the list. It would be interesting to see whether any ER proteins are present among the interactors.
RESPONSE): We have added the p-values to the table rather than the star system. We have added Table 1 that shows all significant interactors with 3xMyc-CARD19 from the experiment that was the source of Figure 3A data. Note that all 5 MIB proteins in our table were part of the 68 significant proteins from this analysis. The unbiased list of proteins sorted by enrichment with p-values is also available in the proteomics results we deposited in the database for open access. We focused our efforts on MICOS, since the MICOS and MIB proteins were significantly enriched across multiple experiments, and involved multiple proteins from the same complex. We have added a sentence to the text clarifying this point.
- The authors show that in the absence of CARD19, mitochondrial inner membrane architecture is altered. While the data are convincing, they only demonstrate swelling / deformation of cristae, not a loss of crista junctions and/or cristae as in MICOS deletion mutants. Therefore, the data do not support the conclusion that CARD19 is a constituent of MICOS (even more so in light of the previous points). This work does not even show that MICOS is altered in the absence of CARD19. Membrane alterations can be the result of dysregulation in any number of processes (many of which take place at organelle contact sites). Thus the authors should limit their conclusion and title to the demonstrated findings that CARD19 interacts with MICOS components (among other proteins?) and has a role in inner membrane architecture.
RESPONSE): The reviewer makes a good point. We have revised our statements and the manuscript title to better reflect the conclusions that can be strongly supported by the data.
Minor points:
- The microscopy images in Fig. 1,2 and SFig. 1,2 are very small and often quite dark, and in many cases it isn't clear immediately to which parts of the image insets correspond. Please modify accordingly. Also for a better comparison an inset of the untreated mitochondria in Fig. 4A would be helpful.
RESPONSE): We have increased the brightness, and in some cases the size, of the images and inset boxes. We have also included insets of the untreated mitochondria in Fig. 4A.
- In Fig. 3B CARD19 is re-expressed into the CARD19-/- cells and claimed to express to a normal WT level. It would be more scientifically correct to show a figure to underline this claim either in the manuscript or in the supplementary files.
RESPONSE): We have added an anti-CARD19 western blot to the supplement that demonstrates similar expression of endogenous CARD19 vs. 3xMyc-WT, 3xMyc-dTM, and 3xMyc-d146 constructs. However, we had difficulty detecting 3xMyc-dCARD and 3xMyc-G73R constructs with the CARD19 antibody (note that the immunogen used for this CARD19 antibody was a large N-terminal peptide encompassing most of the CARD. This peptide is entirely absent in dCARD, and we suspect that G73R greatly diminishes recognition by this antibody, as now indicated in the manuscript text.
- The authors should proofread their manuscript and check that the mentioned figures in the text are linked to the corresponding figures displayed and that figure legends are correct
RESPONSE): Thank you. We believe figure callouts are now entirely accurate.

Round 2
Reviewer 1 Report
Authors addressed my concerns.
Author Response
Thank you. We have performed the requested minor spell check and corrected the few errors that we found.
Reviewer 2 Report
The revised version of the manuscript has been improved and the authors tried to address most concerns raised during the review. However, the alternative protease sensitivity experiment included now does not clarify the topology of the protein and in fact provides even less information than the previously included experiment. Without the PK treatment shown in Fig 1C of the non-revised manuscript, the authors' conclusion is not supported. They should include both experiments in the manuscript.
In addition, while the authors now include p-values for the mass spec data in Fig. 3A, they still do not show an unbiased list of interacting proteins, but limit the table to MICOS and MIB components. This raises concerns about the quality of the data and/or the validity of selecting only these interactors. The authors need to provide an unbiased overview of at least the most prominent interactors, for example in the supplementary files.
Author Response
Thank you for the additional comments. To address the reviewer's first request, we have now added the original ProK assay as a panel added to Supplemental Figure S2 (S2B). With regard to the reviewer's second request, those data were already present in the first revision (please search the manuscript for the callout for Supplemental Table 1; this table was uploaded with our supplemental figures, but it was unfortunately not included in the Supplemental PDF created by Cells - According to the Editors, I needed to submit this as a PDF rather than xlsx format. I have now uploaded a new Supplemental zip file which includes Supplemental Table 1 as a PDF). Supplemental Table 1 includes ALL significant interacting proteins in the proteomics experiment shown in Fig 3A. As is shown in Supplemental Table 1, all 5 of the identified MICOS/MIB interacting partners of Myc-CARD19 in Fig 3A were scored as significant interactions, and these were 5 of 67 total significant interactions identified (the 68th protein on the table is Myc-CARD19, the protein that was immunoprecipitated in the assay).